# Remote Assessment of Parkinson’s Disease Patients Amidst the COVID-19 Lockdown in Mexico

**DOI:** 10.3390/brainsci13071114

**Published:** 2023-07-22

**Authors:** Rodrigo León-García, Emmanuel Ortega-Robles, Oscar Arias-Carrión

**Affiliations:** Unidad de Trastornos del Movimiento y Sueño, Hospital General Dr. Manuel Gea González, Calzada de Tlalpan 4800, Mexico City 14080, Mexico; rodrigo.leong1@gmail.com (R.L.-G.); emmanuel.ortega@salud.gob.mx (E.O.-R.)

**Keywords:** COVID-19, Parkinson’s disease, motor symptoms, non-motor symptoms, telemedicine

## Abstract

The COVID-19 pandemic introduced unprecedented challenges in managing patients with Parkinson’s disease (PD) due to disruptions in healthcare services and the need for social distancing. Understanding the effects of COVID-19 on PD symptoms is crucial for optimizing patient care. We conducted a comprehensive analysis of the data obtained during the period of COVID-19 lockdown, comparing it with analogous timeframes in 2018 and 2019. Our objective was to examine the influence of this unique circumstance on both motor and non-motor symptoms in patients with PD. Telemedicine was employed to assess symptoms using the Movement Disorder Society-sponsored Unified Parkinson’s Disease Rating Scale (MDS-UPDRS). Our findings revealed a notable worsening of symptoms, evidenced by a significant increase in the total MDS-UPDRS score. Specifically, there was an increase in Part III scores, reflecting changes in motor function. However, no differences were observed in Parts I or II, which pertain to non-motor symptoms. Additionally, patient satisfaction and the feasibility of telemedicine consultations were high, highlighting the efficacy of this alternative approach during the pandemic. The COVID-19 pandemic had a discernible impact on PD symptoms, with a significant worsening of motor symptoms observed during the lockdown period. Telemedicine was a valuable tool for remote assessment and follow-up, ensuring continuity of care for individuals with PD in the face of pandemic-related challenges.

## 1. Introduction

Since late December 2019, when the Municipal Health Commission of Wuhan, Hubei Province, China reported the initial cases of a novel severe acute respiratory syndrome called coronavirus disease 2019 (COVID-19), the virus has rapidly spread worldwide. The World Health Organization (WHO) officially declared it a pandemic on 11 March 2020 [1]. In Mexico, the first COVID-19 case was identified on 27 February 2020. On 24 March, the Secretaría de Salud implemented a lockdown in response to the country’s first local infections [2].

Due to the COVID-19 pandemic and the increased risk of severe illness among our patients [3], the Unidad de Trastornos del Movimiento y Sueño at the Hospital General “Dr. Manuel Gea González” in Mexico City suspended in-person consultations on 13 March 2020. Our hospital was designated to attend to COVID-19 patients.

Patients with Parkinson’s disease (PD) require regular follow-up care to manage disease progression. However, the risk of infection at public hospitals in Mexico posed a significant concern for these individuals during the COVID-19 pandemic. Additionally, social isolation and mobility restrictions have been known to affect the mental health of PD patients, exacerbate both motor and non-motor symptoms, and compromise their quality of life [4,5,6]. Consequently, we transitioned to online consultations for patient follow-up, as this provided the safest approach [7]. Several studies have demonstrated the feasibility of online care for PD patients, even when complete motor assessments are not possible. Furthermore, online consultations have been shown to reduce patient travel time and costs while maintaining an equivalent or higher level of patient satisfaction compared to in-person visits [8,9,10,11,12].

However, limited research has been published on the impact of discontinuing in-person care on motor and non-motor symptoms in PD patients. Song et al. reported subjective worsening of both motor and non-motor symptoms assessed by means of a questionnaire, despite no significant changes in the UPDRS Part III score [13]. Similarly, van der Heide et al. observed a decrease in physical activity among patients, which correlated with worsened symptoms such as rigidity, fatigue, tremor, pain, and cognitive impairment, as evaluated by using UPDRS Parts I and II and a subjective symptom-severity survey [14]. Consistent with these findings, Baschi et al. documented the deterioration of motor and non-motor symptoms assessed using UPDRS Parts I and II during the lockdown in Italy [15]. However, a Danish/Swedish cohort study by Hørmann Thomsen et al. showed no significant increase in motor symptoms based on the patient-reported outcome in the Parkinson’s disease scale and the Parkinson KinetiGraph’s bradykinesia median score [16]. Similarly, Kitani-Morii et al. reported a subjective worsening of motor performance among participants, but no significant differences were observed in the scores of PD patients compared to controls [17].

This study aimed to evaluate the feasibility of virtual visits and the satisfaction level of PD patients attending them during the COVID-19 pandemic in Mexico, a developing country where challenges such as digital illiteracy among older and less educated patients, as well as limited access to reliable internet services, could potentially hinder the acceptance of online consultations. Additionally, we aimed to assess any changes in motor and non-motor symptoms during the initial nine months of the COVID-19 lockdown, utilizing the Movement Disorder Society-sponsored Unified Parkinson’s Disease Rating Scale (MDS-UPDRS).

## 2. Materials and Methods

### 2.1. Participants

A total of 160 Parkinson’s disease (PD) patients regularly receiving care at the Unidad de Trastornos del Movimiento y Sueño at the Hospital General “Dr. Manuel Gea González” were included in this study. The diagnosis of PD was based on the clinical diagnostic criteria of the Movement Disorders Society (MDS) [18]. Patients with atypical or secondary parkinsonism and those who had not undergone at least one in-person visit within the 9 months prior to the COVID-19 lockdown (from June 2019 to March 2020) were excluded, resulting in a final sample of 40 patients. One patient who passed away due to causes unrelated to PD or COVID-19 was also excluded, leaving a total of 39 patients who were invited via telephone to schedule an online consultation. It is important to highlight that none of the patients included in our study tested positive for COVID-19 or had any known contact with individuals diagnosed with the disease during the study period. We specifically inquired about COVID-19 symptoms and exposure during the virtual visits with the patients. This information allows us to confidently exclude the direct impact of viral infection on the observed findings.

### 2.2. Virtual Visits

The virtual consultations were conducted by a movement disorders expert from March to December 2020, primarily using the Zoom videoconferencing platform. The recommendations provided by the International Parkinson and Movement Disorder Society for the implementation of telemedicine were followed [19].

Prior to the first virtual visit, patients and their caregivers were given instructions to prepare for the consultation. These instructions included the scheduled consultation time, the use of a device with a camera (smartphone, tablet, or computer), guidance on using the selected videoconferencing platform, preparation of a suitable space for the consultation (quiet, well-lit, preferably close to the internet modem, and with other devices disconnected), availability of a hallway for gait assessment, and the presence of a caregiver or capable assistant if needed. Patients were also requested to provide a phone number with a WhatsApp account and an email address, either their own or that of their caregiver, to receive written instructions.

The virtual visits closely mirrored the in-person visits. They consisted of obtaining the patient’s clinical history, conducting a Parkinson’s-specific examination (including the MDS-UPDRS) omitting rigidity and postural instability testing, and providing treatment recommendations. At the beginning of the consultation, patients were briefly informed about the reasons for conducting the visit virtually, and they were assured that any technical failures would result in rescheduling the consultation. The assistance of the accompanying person was requested if necessary. Neither the examiner nor the patients had prior experience with online consultations. The feasibility of virtual visits was assessed based on the proportion of completed visits as scheduled [20].

### 2.3. Feasibility and Acceptability Survey

In mid-December 2020, patients who had completed at least one virtual visit were contacted via phone to participate in a survey assessing their experiences with the online consultations. The survey questions were based on the patient questionnaire developed by Hanson et al. [21] and the modified version by Shalash et al. [22]. The questions utilized a 10-point Likert-scale format, where a score of 1 indicated “strongly disagree” or “least satisfied” and a score of 10 indicated “strongly agree” or “most satisfied” (Appendix A). The questions were grouped into three categories: patient satisfaction (questions 6, 7, 8, 14, 15, 17, 18, and 19), set-up or preparation (questions 3 and 4), and quality of service (questions 9, 10, 11, and 12). To rate each category, the median of the responses to the questions within the respective domain was calculated [21]. Additionally, patients were asked to provide information on the number of virtual visits they completed, the type of device used for the visits, the time required for a regular in-person visit, and the time spent preparing for the virtual visit, and they were allowed to provide general impressions about telemedicine through an open-ended question. The survey was conducted using the Google Forms tool to ensure anonymous responses.

### 2.4. Assessment of Motor and Non-Motor Symptoms

In order to evaluate the impact of the COVID-19 lockdown on the health of PD patients, we conducted a comprehensive analysis of their motor and non-motor symptoms. The assessment was based on the scores obtained from the MDS-UPDRS during three distinct time periods. The study comprised two nine-month intervals of pre-pandemic in-person consultations (T1 and T2) and a nine-month period of virtual consultations during the confinement period (T3), as illustrated in Figure 1. Initially, we compared the UPDRS scores between the two in-person visit periods (T1 vs. T2) to estimate changes associated with the natural progression of the disease over time. Subsequently, we compared the outcomes measured during the immediate period prior to the pandemic restrictions with those obtained during the lockdown period (T2 vs. T3). This comparison allowed us to assess the changes produced by both the disease course and any additional aggravating factors associated with the pandemic situation. In cases where patients had multiple assessments within a given period, we calculated the median scores to ensure representative values. Our analysis encompassed the total UPDRS scores as well as segmentations into Part I, Part II, Part III, and the Hoehn and Yahr scale. Moreover, within Part III, further subdivisions were made, including upper limbs (UL), lower limbs (LL), right side (RS), and left side (LS). Notably, scores for rigidity and postural instability testing were not included, as their assessment inherently requires physical intervention from the assessor. By employing these comprehensive evaluations, we aimed to capture the full spectrum of motor and non-motor symptoms experienced by PD patients during the pandemic. We excluded certain items from analysis to ensure the validity and reliability of our results [23].

### 2.5. Statistical Analyses

To investigate differences in the UPDRS scores between periods of in-person and virtual visits, we employed a mixed effects model for repeated measures data. This model incorporated the periods of time as the fixed effect factor and the subjects as the random effect. Pairwise post hoc testing was conducted using Tukey’s method.

All statistical analyses were performed using GraphPad Prism version 8.0.1 software, with a significance level set at 0.05. The software utilized a compound symmetry covariance matrix for the mixed model and implemented the restricted maximum likelihood (REML) method to estimate the model parameters.

It is important to note that among the 24 patients who completed at least one virtual visit, five patients did not have an evaluation during the T1 period.

## 3. Results

### 3.1. Feasibility and Acceptability of Online Consultations

The feasibility of online consultations was high, although not all PD patients expressed interest in participating. Thirteen out of the 39 patients declined to proceed with telemedicine, citing stability and satisfaction with telephone consultations as sufficient. Among the remaining 26 patients (66.6%) who agreed to virtual visits, two of them had to cancel their appointments twice for unknown reasons and could not be re-contacted. Ultimately, 24 patients completed the scheduled online visits, resulting in a feasibility rate of 92.3%. In summary, 24 out of the 39 patients (61.5%) actively participated in the virtual consultations (Figure 2). Some patients who completed their initial virtual visit continued to receive care in this format. From March to December 2020, a total of 40 virtual visits were successfully conducted, with a median of 2 (range: 1–3) visits per patient (Table 1).

The survey results indicated that virtual consultations were well received by PD patients. Out of the 24 patients who underwent at least one virtual consultation, 20 of them (83.3%) responded to the survey. The most commonly used device for teleconsultations was a laptop (50%), followed by a smartphone (35%) and a desktop computer (15%). Patient responses demonstrated a high level of satisfaction with online consultations, with a median satisfaction score of 9.5 (range: 6.5–10), a median score of 10 (range: 8–10) for set-up/preparation, and 10 (range: 10–10) for quality of service. All patients indicated that they found the virtual consultation service valuable.

### 3.2. Personal Connection and Time Savings

Regarding the personal connection established between patients and the specialist during virtual visits, 75% of respondents felt it was equivalent to an in-person visit, while 20% perceived it as less personal. A small percentage (5%) reported feeling a stronger connection, and none reported a complete absence of connection. The average preparation time for virtual visits was 16.1 min, ranging from 2 to 45 min. On average, patients saved 123.7 min in consultation time compared to an in-person visit, with variations ranging from 15 to 400 min.

### 3.3. Impact of COVID-19 Lockdown on PD Patients

Our online consultations revealed a notable worsening in motor symptoms during the pandemic period that could not be attributed to the natural progression of the disease. There were no statistically significant differences in the scores of the MDS-UPDRS scale between the two in-person visit periods (T1 and T2), either in the total score or when they were analyzed separately (Figure 3). However, a significant average increase of 6.86 points (95% CI: 0.85–12.88, *p* = 0.0232) was observed in the total UPDRS score, primarily driven by a 6.54-point increase (95% CI: 2.64–10.44, *p* = 0.0010) in the clinician-scored motor evaluation, UPDRS Part III. No significant differences were found in the scores of UPDRS Parts I and II. A subgroup analysis of Part III revealed significant changes in the upper limbs (mean: 3.63, 95% CI: 1.51–5.74, *p* = 0.0008), right side (mean: 1.65, 95% CI: 0.21–3.08, *p* = 0.0227), and left side (mean: 2.70, 95% CI: 0.97–4.43, *p* = 0.0020), but not in the lower limbs. Additionally, a small but significant increase in the mean Hoehn and Yahr score of 0.24 (95% CI: 0.47–0.01, *p* = 0.0439) was detected.

## 4. Discussion

Our findings highlight the value and high acceptance of virtual visits among our clinic patients, aligning with previous studies [9,21,22]. The feasibility of this modality was excellent, and patient satisfaction and perceived quality of care were comparable to in-person visits. Moreover, virtual visits offered significant time savings for patients and potential cost savings by eliminating expenses such as transportation fees, fuel, and parking.

Although the physician conducting the virtual visits reported an increase in symptoms of anxiety and depression, as well as reduced physical activity among some patients during the confinement, these changes were not reflected in the non-motor symptoms assessed by UPDRS Part I. However, it is noteworthy that 58.3% of the patients required medication adjustments for non-motor symptoms, indicating the importance of addressing these aspects during virtual consultations (Table 1). Previous studies, including Falla et al. [24] and Shalash et al. [25], have also reported a deterioration in the non-motor aspects of daily life, as measured by UPDRS Part I. It is important to note that these findings may vary across different populations and countries, reflecting the unique experiences and conditions of each study population during the pandemic. For instance, Falla et al. focused on the initial two-month period of the pandemic lockdown in Italy. During this period, Italy faced significant stress and uncertainty, and was heavily impacted by the pandemic [26]. The specific circumstances in Italy at that time, such as increased stress levels and the burden of the pandemic, may have contributed to the observed worsening in non-motor aspects of daily life. It is crucial to contextualize the findings of these studies within the unique circumstances of each population and country. Factors such as the severity of the pandemic outbreak, the healthcare system’s response, and individual experiences during the lockdown can influence the observed outcomes.

The impact of the COVID-19 lockdown on individuals with PD became evident through the observed worsening of motor symptoms during the pandemic period, independent of the disease’s natural course. This deterioration was reflected in an increase in the total Unified Parkinson’s Disease Rating Scale (UPDRS) score, primarily driven by significant changes in motor evaluation, specifically in UPDRS Part III. Importantly, the increase in the total UPDRS score exceeded the threshold for minimal clinically important difference, which is set at 4.63 points [27]. Previous studies have consistently shown that over a 9-month period, one would not expect substantial changes in motor symptoms among individuals with PD due to the disease’s natural history [28,29,30]. This was reflected in the comparison between the T1 and T2 periods, where the mean increase was 5.62% (95% CI: −5.29–16.52%). However, the increase observed between the T2 and T3 periods was remarkable, with a mean increase of 20.79% (95% CI: 10.55–31.02%). This significant increase suggests that the changes in motor symptoms were specifically related to the pandemic lockdown. Notably, significant changes were observed in the upper limbs, right side and left side, while no significant differences were found in the lower limbs. Furthermore, a small but significant increase in the mean Hoehn and Yahr score was detected, indicating a slight progression in disease severity. These findings collectively highlight the notable impact of the pandemic lockdown on the motor symptoms and disease progression in individuals with PD.

Previous studies have reported the worsening of motor symptoms in individuals with PD during the confinement period, as evident from subjective questionnaires [13,14,15,17,25,31,32,33,34,35,36,37]. However, several studies that evaluated the UPDRS Part III did not find significant changes between the pandemic and pre-pandemic periods [13,24], or the changes were only marginally significant [31]. It is important to note that these studies had shorter evaluation periods ranging from 1 to 3 months. In contrast, our study spanned a longer duration of nine months, allowing for more pronounced changes to emerge. Additionally, our study included individuals with more advanced disease stages, with 25% of the subjects classified as Hoehn and Yahr stages > 3. Furthermore, medication adjustments were required for 50% of our patients to manage both motor and non-motor symptoms, as previously mentioned. Another study by Hørmann Thomsen et al. [16] did not identify significant differences in motor symptoms during the COVID-19 lockdown, but their evaluation period was also shorter (3 months), and the living conditions of the study population (Denmark and Sweden) differ significantly from those of our patients [38]. In a study by Pauly et al. [39], no changes in the UPDRS Part III were observed despite the long evaluation period of one year. Unfortunately, demographic data for the sample in that study were not provided, making it difficult to determine the reason for the lack of detectable motor impairment. In contrast, studies by Ineichen et al. [32], Shalash et al. [25], and Capecci et al. [34], which had evaluation periods longer than six months, yielded results consistent with our findings. These studies reported an increase in the UPDRS Part III score during the lockdown period. Therefore, the duration of the evaluation period appears to be an important factor in assessing motor deterioration using the MDS-UPDRS scale. Taken together, these findings underscore the importance of considering the evaluation period’s duration when assessing changes in motor symptoms using the MDS-UPDRS scale in individuals with PD.

Additionally, we observed no significant differences in UPDRS Part II scores, which contrasts with findings from previous studies [14,15,34]. Several hypotheses have been proposed to explain the worsening of motor symptoms during the confinement period. The stress and anxiety stemming from uncertainties surrounding the pandemic’s health and economic impact, coupled with the measures of social distancing and isolation, may have contributed to the deterioration of both motor and non-motor symptoms in individuals with PD [25,33,40,41]. Stress alone has been shown to exacerbate Parkinson’s symptoms [5], and while some PD patients attempted to maintain physical activities at home, recent evidence suggests that structured and supervised exercise routines are necessary to prevent symptom deterioration [42]. The decrease in physical activity and drastic changes in daily habits brought about by the lockdown measures may have further compounded the impact on motor symptoms in this population.

We acknowledge several limitations in our study. The small sample size may have limited our power to detect longitudinal differences, particularly in the assessment of UPDRS Parts I and II. The sample size was constrained by the number of PD patients who agreed to participate in telemedicine follow-ups, which may introduce bias. Furthermore, our sample, consisting of patients from various socioeconomic and educational backgrounds in Mexico City, may not be representative of the entire country’s population. The acceptance and feasibility of telemedicine may vary in different regions. Additionally, we did not stratify subjects based on disease progression levels, preventing us from concluding the varying effects on patients with different disease durations. These limitations could be addressed in the future by means of a multicenter study design.

While Part I of the UPDRS scale is a valid tool for assessing non-motor symptoms of PD [43], our study did not apply other specific scales to measure anxiety, depression, stress, sleep, or other non-motor symptoms. Incorporating such scales could provide a more comprehensive and validated understanding of individual symptoms. Although remote assessment during the pandemic presented challenges, such as the inability to evaluate postural rigidity and stability remotely (resulting in six missing values), the comparison between in-person and virtual visits remains valid as these items were not considered in both periods [23].

Despite the acknowledged limitations, our study provides valuable insights into the importance of providing special attention and appropriate disease management for individuals with PD during periods of social confinement, in order to prevent the exacerbation of symptoms. It is crucial to conduct thorough evaluations of both motor and non-motor symptoms and implement well-planned, supervised exercise programs. Online consultations have emerged as a valuable tool for conducting follow-up assessments, particularly during extraordinary situations such as the COVID-19 pandemic. They offer a means to manage patients who are unable to visit the clinic in person due to various reasons. These virtual consultations provide an alternative approach to ensure continuous care and support for PD patients, facilitating ongoing monitoring and intervention, especially during periods of restricted physical access. The findings of our study emphasize the importance of adapting healthcare strategies to address the specific needs of PD patients during times of crisis and social confinement. By implementing comprehensive and tailored management plans, healthcare providers can optimize patient outcomes and enhance the overall well-being of individuals with PD.

## 5. Conclusions

To the best of our knowledge, this is the first study conducted in Mexico to assess the feasibility of virtual visits for PD during the COVID-19 pandemic. Additionally, our study is the first in our country to objectively examine the changes in PD patients between the pre-pandemic and pandemic periods using the UPDRS assessment. By providing important data, our study contributes to the growing body of knowledge in this field and emphasizes the significance of addressing the unique challenges faced by PD patients during unprecedented times.

Our findings highlight the value of virtual visits as an alternative approach to PD management. The high level of patient satisfaction, time savings, and cost-effectiveness associated with virtual care, along with its ability to detect and address motor symptom deterioration, make it a valuable tool in providing comprehensive care for PD patients. Future studies should further explore the potential benefits and optimize the implementation of virtual visits to enhance patient outcomes and improve healthcare for individuals with PD.

## Figures and Tables

**Figure 1 brainsci-13-01114-f001:**
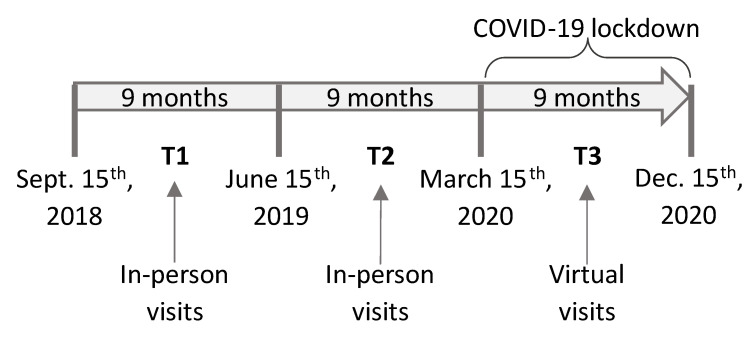
Timeline of in-person and virtual visit periods. T1 represents the normal in-person visits; T2, the pre-pandemic in-person visits; and T3, the virtual visits during the COVID-19 pandemic.

**Figure 2 brainsci-13-01114-f002:**
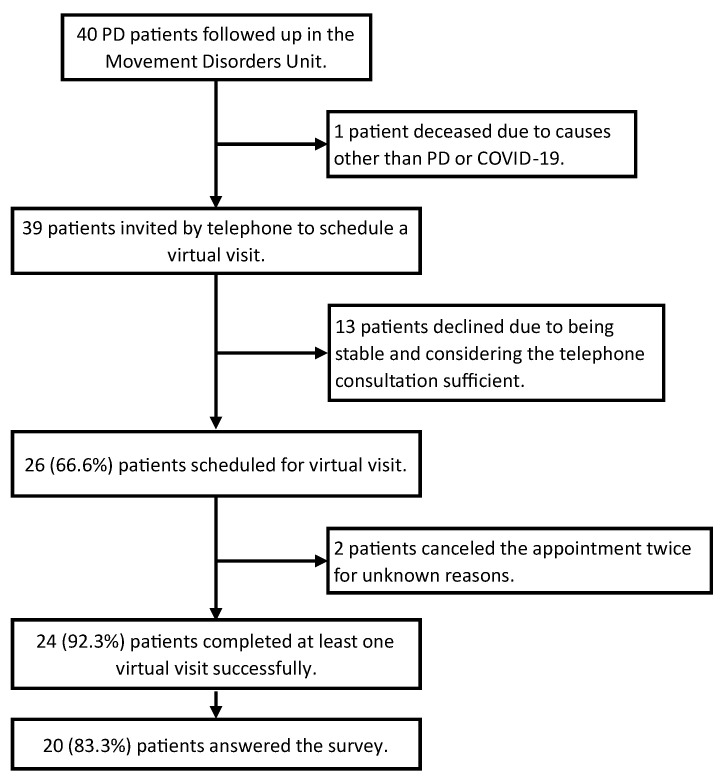
Flowchart of Parkinson’s disease Patient Recruitment for Virtual Visits and Satisfaction Survey.

**Figure 3 brainsci-13-01114-f003:**
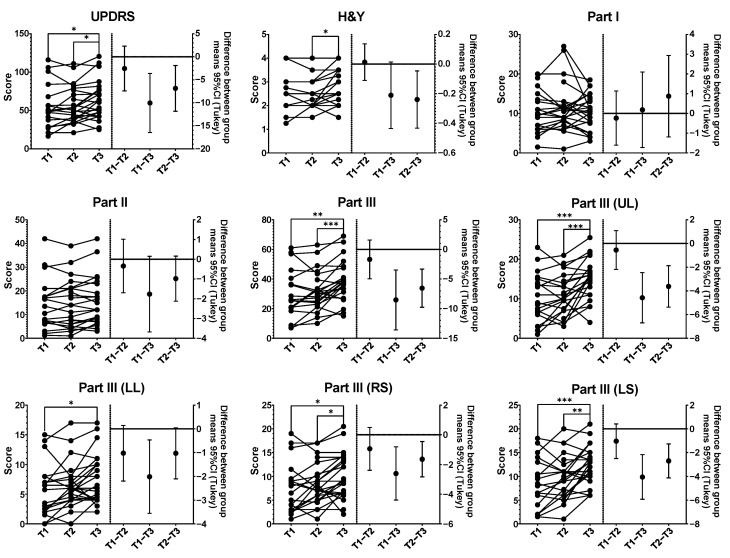
Comparison of UPDRS scores between T1 (in-person visits), T2 (in-person visits), and T3 (virtual visits during lockdown). UPDRS: Movement Disorder Society-sponsored Unified Parkinson’s Disease Rating Scale; H&Y: Hoehn and Yahr scale; Part I: non-motor aspects of experiences of daily living; Part II: motor aspects of experiences of daily living; Part III: motor examination; UL: upper limbs; LL: lower limbs; RS: right side; LS: left side. In all graphical representations, a before–after plot is employed to illustrate the scores at each period (T1, T2, and T3) on the left side. On the right side, the mean difference of scores between periods is displayed, accompanied by the corresponding 95% confidence interval (CI). The statistical analyses utilized a mixed effects model, and Tukey’s post hoc test was applied to assess the significance of observed differences. Significance levels are denoted as * (*p* < 0.05), ** (*p* < 0.01), and *** (*p* < 0.001).

**Table 1 brainsci-13-01114-t001:** Clinical characteristics of patients and outcomes of the telemedicine consultations.

	Mean/N°/Median	SD/Frequency	Range
**Demographics**			
Age (years)	69.96	6.61	58–83
Sex (female/male)	12/12	50%/50%	
Age of onset (years)	60.42	8.94	47–75
Duration of illness (years)	9.54	5.36	2–20
MDS-UPDRS ^1^	59.50	24.13	34.5–106
Hoehn and Yahr scale ^1^	2.65	0.77	1.5–4
Use of dopaminergic medication ^1^	24	100.0%	
Use of Levodopa ^1^	21	87.5%	
Highest level of education			
Illiterate	2	8.3%	
Elementary school	7	29.2%	
Middle school	3	12.5%	
High school	1	4.2%	
University	11	45.8%	
**Outcomes**			
Number of virtual visits (median Q1)	2		1–3
Satisfaction (median Q6, 7, 8, 14, 15, 17, 18, 19)	9.5		6.5–10
Setup/preparation (median Q3)	10		8–10
Quality of service (median Q9, 10, 11, 12)	10		10–10
Setup time for virtual visits (minutes, Q4)	16.10	11.63	2–45
Saved time for patients (minutes, Q4, 5)	123.65	83.02	15–400
Device used for the virtual visit			
Desktop	3	15.0%	
Smartphone	7	35.0%	
Laptop	10	50.0%	
Tablet	0	0.0%	
Number of evaluations in the period (median)			
T1	2		0–3
T2	3		1–5
T3	2		1–3
Needed dopaminergic medication adjustment	12	50.0%	
Needed medication adjustment for non-motor symptoms	14	58.3%	

^1^ Evaluated or retrieved in the last in-person visit.

## Data Availability

The data presented in this study are available on request from the corresponding author upon reasonable request.

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
