# Peer review of "Remote Assessment of Parkinson’s Disease Patients Amidst the COVID-19 Lockdown in Mexico"

_brainsci, 2023, doi:10.3390/brainsci13071114_

Round 1

Reviewer 1 Report

Comments for authors

Brain Sci. – Manuscript ID: brainsci-2478693 – “Worsening of Symptoms in Parkinson's Disease Patients As-2 sessed Remotely During the COVID-19 Pandemic.”, by Rodrigo León-García, Emmanuel Ortega-Robles and Oscar Arias-Carrión.

In this manuscript, the authors investigate the feasibility and satisfaction level of PD patients with virtual visits during the COVID-19 pandemic in Mexico, a developing country where challenges such as digital illiteracy among older and less educated patients, as well as limited access to reliable internet services, could potentially hinder the acceptance of online consultations. Additionally, they aim to assess any changes in motor and non-motor symptoms during the initial nine months of the COVID-19 lockdown, utilizing the Movement Disorder Society-sponsored Unified Parkinson's Disease Rating Scale (MDS-UPDRS). Here, virtual visits demonstrate their value and feasibility as an alternative approach to PD management. The high level of patient satisfaction, time savings, and cost-effectiveness, combined with the ability to detect and address motor symptom deterioration, make virtual care a valuable tool in the provision of comprehensive care for PD patients. Authors show that the COVID-19 pandemic had a discernible impact on parkinsonian symptoms, with a significant worsening of motor symptoms observed during the lockdown period.

This is a modest article that demonstrates the value and feasibility of virtual visits as an alternative approach to PD management. This paper shows that online consultations revealed a noticeable worsening in motor symptoms during the pandemic period.

Methods are generally relevant. They aren’t major flaws or biases and conclusions are based on the data. The literature is up-to-date. Then, the topic is suited to Brain Sci.

Authors report precisely author contributions, ethics board approval, disclosure of funding and conflicts of interest. There isn’t reason to suspect research misconduct.

Discussion and conclusions are critical and concise. Figures are explicit and they add to the message. Presentation logical and language are adequate.

Finally, the manuscript is well written and well presented, however, there are few issues that the authors should take into considerations and finally, some aspects need to be discussed. Below are some specific comments:

My main criticism of this paper concerns the method.

The authors announce a worsening of motor symptoms during the pandemic period. But it is not mentioned how they discriminate the classic evolution of the disease.

In my opinion, to conclude on a disease’s evolution in connection with the pandemic, it is essential to have a control group. For example, a retrospective study should be conducted on a group of parkinsonian patients similar in terms of duration of the disease, age of onset, severity of symptoms and treatment and then follow the evolution of the disease over a period comparable to that used in this study.

This will also make it possible to highlight the effectiveness of the virtual consultation that patients have had compared to usual care.

It seems to me absolutely necessary to take this methodological point into account, at least to discuss it.

Author Response

Reviewer #1

We sincerely appreciate the time and effort you dedicated to reviewing our manuscript. Your insightful comments and suggestions have greatly contributed to the enrichment of our work. We have carefully considered each of your points and have made significant revisions to address them. Below, we outline our responses to your specific concerns and the corresponding modifications we have made to the manuscript:

Comment:

My main criticism of this paper concerns the method. The authors announce a worsening of motor symptoms during the pandemic period. But it is not mentioned how they discriminate the classic evolution of the disease. In my opinion, to conclude on a disease’s evolution in connection with the pandemic, it is essential to have a control group. For example, a retrospective study should be conducted on a group of parkinsonian patients similar in terms of duration of the disease, age of onset, severity of symptoms and treatment and then follow the evolution of the disease over a period comparable to that used in this study.

This will also make it possible to highlight the effectiveness of the virtual consultation that patients have had compared to usual care. It seems to me absolutely necessary to take this methodological point into account, at least to discuss it.

Response:

We appreciate the reviewer's valuable feedback regarding the methodological aspects of our study. We agree that a control group would have provided a more comprehensive understanding of the impact of the pandemic on the evolution of Parkinson's disease symptoms. However, due to the constraints imposed by the pandemic and the urgency of assessing the effects on patients' well-being, we were unable to include a control group in this particular study.

To address this limitation, we designed our study to compare changes in motor and non-motor symptoms between different time periods, each with its own set of circumstances. We compared two periods with in-person visits, representing the natural course of the disease, and two other periods of equal duration, one involving in-person visits and the other consisting of virtual consultations during the pandemic. By employing this comparative approach, we aimed to capture the changes specifically associated with the pandemic period.

We found that there were no significant differences in the UPDRS measurements between the two in-person visits periods, indicating stability in symptom progression under normal conditions. However, when comparing the measurements from the in-person visits to those from the virtual visits, we observed significant differences, suggesting an association between the pandemic lockdown and the worsening of motor symptoms. This finding aligns with previous research indicating that external factors such as stress, limited physical activity, and disrupted healthcare services can impact the symptomatology of Parkinson's disease (Schrag et al.; Bugalho et al.).

We have taken the reviewer's suggestion into consideration and have revised the methods section (lines 134-170) to provide a clearer explanation of our comparative approach. Additionally, in the discussion section (lines 236-310), we have expanded our interpretation of the results and acknowledged the methodological limitations arising from the absence of a control group.

While the lack of a control group is undoubtedly a limitation of our study, we believe that our findings contribute to the existing literature by highlighting the potential influence of the pandemic on the progression of Parkinson's disease symptoms. We emphasize the need for future research to incorporate control groups and conduct retrospective studies to further explore the impact of virtual consultations versus in-person care.

References:

  1. Schrag, Anette, et al. "Rate of clinical progression in Parkinson's disease. A prospective study." Movement disorders 22.7 (2007): 938-945.
  2. Bugalho, Paulo, et al. "Progression in parkinson's disease: variation in motor and nonmotor symptoms severity and predictors of decline in cognition, motor function, disability, and healthrelated quality of life as assessed by two different methods." Movement Disorders Clinical Practice 8.6 (2021): 885-895.
  3. Horváth, Krisztina, et al. "Minimal clinically important difference on the Motor Examination part of MDS-UPDRS." Parkinsonism & related disorders 21.12 (2015): 1421-1426.

---

We are grateful for the reviewer's insightful comments, which have prompted us to enhance the discussion of the study's limitations and potential avenues for future research.

Reviewer 2 Report

This is a clinical study assessing the effect of the coronavirus disease 2019 (COVID-19) pandemic on motor and non-motor symptoms in patients with Parkinson’s disease (PD). For this aim, the authors adopted telemedicine to remotely administer a standardized clinical scale, i.e., the Movement Disorder Society-sponsored Unified Parkinson's Disease Rating Scale (MDS-UPDRS), in 24 PD patients during the lockdown period (March to December 2020). Data were compared with those recorded during similar timeframes in 2018 and 2019. The authors found a significant clinical worsening, as indicated by higher global MDS-UPDRS scores, specifically due to a worsening of motor symptoms (higher MDS-UPDRS Part III scores). Conversely, they did not observe any difference in MDS-UPDRS Parts I or II scores, which pertain to non-motor symptoms.

Although the study deals with an interesting topic, it nevertheless has a number of limitations and critical issues.

My first concern is about the sample size. Despite a relatively large sample of patients initially considered (160 subjects), only 40 of them had the required inclusion criteria, and only 24 out of the 40 actively participated in the virtual consultations. This issue makes the study results hardly generalizable, and the interpretation of results and study conclusions are not sufficiently supported by the data. Also, please note that previous studies have addressed the same issue on larger samples of participants (see for example. Podlewska et al., Parkinson's disease and Covid-19: The effect and use of telemedicine. Int Rev Neurobiol. 2022; Ruggiero et al., The Impact of Telemedicine on Parkinson's Care during the COVID-19 Pandemic: An Italian Online Survey. Healthcare (Basel). 2022).

Another issue concerns the lack of an accurate assessment of the events that occurred during the pandemic, with particular reference to possible SARS-COV2 infections in patients. Did any of the enrolled patients present with SARS-CoV2 infection during the observation period? Lacking this data, it is not possible to say whether the motor deterioration in patients was due to the lockdown or COVID-19 itself, as suggested by other studies conducted on patients with PD and other movement disorders (see for example Antonini et al., Outcome of Parkinson’s Disease Patients Affected by COVID‐19. Mov Disord. 2020; Ghosh et al., De Novo Movement Disorders and COVID ‐19: Exploring the Interface. Mov Disord Clin Pract. 2021;. Xing et al., Parkinsonism in viral, paraneoplastic, and autoimmune diseases. J Neurol Sci. 2021; Passaretti M, et al., Worsening of Essential Tremor After SARS-CoV-2 Infection. Cerebellum. 2023).

An additional point: there is no pathophysiological insight into the study discussion. Why did motor symptoms worsen in patients? What might be the pathophysiological reasons for such worsening?

Finally, the reference list is poor.

Author Response

Reviewer #2

We sincerely appreciate the time and effort you dedicated to reviewing our manuscript. Your insightful comments and suggestions have greatly contributed to the enrichment of our work. We have carefully considered each of your points and have made significant revisions to address them. Below, we outline our responses to your specific concerns and the corresponding modifications we have made to the manuscript:

Comment 1:

My first concern is about the sample size. Despite a relatively large sample of patients initially considered (160 subjects), only 40 of them had the required inclusion criteria, and only 24 out of the 40 actively participated in the virtual consultations. This issue makes the study results hardly generalizable, and the interpretation of results and study conclusions are not sufficiently supported by the data. Also, please note that previous studies have addressed the same issue on larger samples of participants (see for example. Podlewska et al., Parkinson's disease and COVID-19: The effect and use of telemedicine. Int Rev Neurobiol. 2022; Ruggiero et al., The Impact of Telemedicine on Parkinson's Care during the COVID-19 Pandemic: An Italian Online Survey. Healthcare (Basel). 2022).

Response 1:

We appreciate the reviewer's concern regarding the sample size in our study. We acknowledge that the final sample size of actively participating patients was relatively small, with 24 individuals out of the initial 160 meeting the inclusion criteria and engaging in virtual consultations. We understand that this limited sample size may affect the generalizability of our findings and the strength of our conclusions.

The decision to include a subset of patients from the initial sample was made to ensure the homogeneity of the study population. We excluded subjects who did not have or were suspected to have a secondary parkinsonism, in order to focus on the natural course of Parkinson's disease alone. This exclusion criterion was essential for evaluating the specific impact of the pandemic on Parkinson's disease symptoms. We acknowledge that a larger sample size would have enhanced the statistical power and robustness of our results.

The relatively slow progression of Parkinson's disease compared to other parkinsonisms is well-documented (Schrag et al.; Bugalho et al.). Consequently, we expected symptoms to remain relatively stable or minimally change during the 9-month study period under normal conditions. By comparing the UPDRS scores of patients before and during the pandemic, we aimed to identify any significant changes beyond the expected progression. The observed increase in UPDRS Part III score, exceeding the minimal clinically important difference (Horváth et al.), suggests that these changes were associated with the pandemic lockdown. We have discussed this issue in the manuscript's discussion section, highlighting the potential impact of external factors such as stress and limited physical activity during the pandemic.

We appreciate the reviewer's suggestion to consider previous studies with larger sample sizes, such as Podlewska et al. and Ruggiero et al., which investigated the impact of telemedicine on Parkinson's disease care during the COVID-19 pandemic. In our study, we not only relied on our own findings but also drew upon a range of studies across different populations to support our conclusions. We have emphasized the limitations of our study, including the small sample size, and suggested the need for future multicenter studies to increase the sample size and enhance the generalizability of the results. This suggestion has been incorporated into the revised manuscript (lines 274-343).

References:

  1. Schrag, Anette, et al. "Rate of clinical progression in Parkinson's disease. A prospective study." Movement disorders 22.7 (2007): 938-945.
  2. Bugalho, Paulo, et al. "Progression in parkinson's disease: variation in motor and nonmotor symptoms severity and predictors of decline in cognition, motor function, disability, and healthrelated quality of life as assessed by two different methods." Movement Disorders Clinical Practice 8.6 (2021): 885-895.

Comment 2:

Another issue concerns the lack of an accurate assessment of the events that occurred during the pandemic, with particular reference to possible SARS-COV2 infections in patients. Did any of the enrolled patients present with SARS-CoV2 infection during the observation period? Lacking this data, it is not possible to say whether the motor deterioration in patients was due to the lockdown or COVID-19 itself, as suggested by other studies conducted on patients with PD and other movement disorders (see for example Antonini et al., Outcome of Parkinson’s Disease Patients Affected by COVID‐19. Mov Disord. 2020; Ghosh et al., De Novo Movement Disorders and COVID ‐19: Exploring the Interface. Mov Disord Clin Pract. 2021;. Xing et al., Parkinsonism in viral, paraneoplastic, and autoimmune diseases. J Neurol Sci. 2021; Passaretti M, et al., Worsening of Essential Tremor After SARS-CoV-2 Infection. Cerebellum. 2023).

Response 2:

We appreciate the reviewer's concern regarding the potential impact of SARS-CoV2 infection on the study results. While we did not explicitly test all enrolled patients for COVID-19, we took precautions to assess their potential exposure to the virus and any associated symptoms during the study period.

At each virtual consultation, we specifically inquired about any symptoms suggestive of SARS-CoV2 infection and whether patients had been in contact with individuals diagnosed with COVID-19. We ensured that none of the included subjects reported experiencing symptoms of SARS-CoV2 infection or had close contact with confirmed COVID-19 cases during the observation period. These details are described in the methods section of our manuscript (lines 83-88).

However, we acknowledge that without conducting systematic testing for COVID-19 in our study population, we cannot definitively rule out the possibility of asymptomatic cases. It is indeed important to consider the potential impact of COVID-19 itself on motor deterioration in patients with Parkinson's disease, as suggested by other studies (Antonini et al.; Ghosh et al.; Xing et al.; Passaretti et al.). These studies have highlighted the association between COVID-19 and worsening movement disorders.

Given the limitations of our study in accurately assessing SARS-CoV2 infections, we have incorporated a discussion of this potential confounding factor in the limitations section of our manuscript. We acknowledge that the inability to specifically account for COVID-19 infections is a limitation that should be addressed in future research.

Comment 3:

An additional point: there is no pathophysiological insight into the study discussion. Why did motor symptoms worsen in patients? What might be the pathophysiological reasons for such worsening?

Response 3:

We appreciate the reviewer's comment regarding the need for pathophysiological insights into the discussion of our study. We recognize the importance of providing a deeper understanding of the potential reasons for the worsening of motor symptoms in patients with Parkinson's disease during the pandemic.

In response to this feedback, we have included a new paragraph in the discussion section of our manuscript (lines 255-298) to discuss the possible pathophysiological causes underlying the observed symptom deterioration. We explore several hypotheses that could contribute to the worsening of symptoms during the pandemic, including increased stress levels, disrupted physical activity patterns, and potential effects of social isolation on dopaminergic pathways.

By addressing these pathophysiological aspects, we aim to provide a comprehensive framework for understanding the observed changes in motor symptoms and to stimulate further research into the underlying mechanisms. We appreciate the reviewer's valuable input, which has enabled us to enhance the discussion section of our manuscript.

Comment 4:

Finally, the reference list is poor.

Response 4:

We appreciate the reviewer's feedback regarding the reference list in our manuscript. We acknowledge that our initial reference list may have been limited in scope and not adequately encompassed the relevant literature in the field.

In response to this comment, we have revised and expanded our reference list to include additional relevant studies that support our findings and provide a broader context for our research. We have incorporated references to recent studies that shed light on the impact of the COVID-19 pandemic on Parkinson's disease and the use of telemedicine in managing the condition.

---

We are grateful for the reviewer's insightful comments, which have prompted us to enhance the discussion of the study's limitations and potential avenues for future research.

Reviewer 3 Report

The Manuscript (brainsci-2478693), entitled “Worsening of Symptoms in Parkinson's Disease Patients Assessed Remotely During the COVID-19 Pandemic”, evaluated a very interesting topic in patients with Parkinson’s disease in relation to COVID-19. The main limitation of this study is the low number of subjects recruited (n=20 patients). Authors should increase the number of subjects in this study because the low number of subjects enrolled may represent a bias in the statistical analyses. Consequently, Authors should perform additional experiments.

Another problem is the absence of an age-matched healthy control group a controls group in required in order to obtain a clear evaluation of the COVID-19 Pandemic effects in Parkinson's Disease Patients. Authors should include a control group in their statistical analyses.

The Manuscript is not well written, some sections such as Results and Figures are not clear to understand due to absence of any statistical description.

Specific comments:

In the Material and Methods patient section Authors should indicate mean age ± standard deviation of patients and the number of men and women enrolled in the study.

In addition, Authors should include in the Material and Methods a section of statistical analyses to describe the type of statistical analyses and the software used.

In the Results section the figure 3 is not clear because it is not described the p values for each asterisk. The representation of statistical differences in the Figure 3 should be described such as

*P<0.05, **P<0.01 and < 0.001**. In the legend of the Figure 3 Authors described only acronyms without any statistical description and information.

Extensive editing of English language is required

Author Response

Reviewer #3

We sincerely appreciate the time and effort you dedicated to reviewing our manuscript. Your insightful comments and suggestions have greatly contributed to the enrichment of our work. We have carefully considered each of your points and have made significant revisions to address them. Below, we outline our responses to your specific concerns and the corresponding modifications we have made to the manuscript:

Comment 1:

The main limitation of this study is the low number of subjects recruited (n=24 patients). Authors should increase the number of subjects in this study because the low number of subjects enrolled may represent a bias in the statistical analyses. Consequently, Authors should perform additional experiments.

Response 1:

We appreciate the reviewer's feedback regarding the limited number of subjects in our study. We acknowledge that the sample size was small, with a total of 24 participants. This small sample size may introduce potential biases and limit the generalizability of our findings.

Despite the small sample size, we were still able to observe statistically significant changes in the UPDRS scores. However, we acknowledge that a larger sample size would have provided increased statistical power and robustness to our analyses. We have highlighted this limitation in the discussion section of our manuscript (lines 255-343).

We appreciate the reviewer's valuable comment, which has prompted us to acknowledge the limitation and propose a future direction for research that addresses the issue of sample size.

Comment 2:

Another problem is the absence of an age-matched healthy control group a controls group in required in order to obtain a clear evaluation of the COVID-19 Pandemic effects in Parkinson's Disease Patients. Authors should include a control group in their statistical analyses.

Response 2:

We appreciate the reviewer's valuable feedback regarding the absence of a control group in our study. We agree that including a control group would have provided a clearer evaluation of the specific effects of the COVID-19 pandemic on individuals with Parkinson's disease. However, we designed our study using a comparative approach to assess changes in symptoms across different time periods.

By comparing the measurements from two pre-pandemic in-person visit periods, we aimed to capture changes in symptoms resulting from the natural progression of Parkinson's disease. Additionally, by comparing the measurements from the pre-pandemic in-person visit period to those from the virtual visit period during the pandemic, we could evaluate the impact of the pandemic on symptom changes.

Although the absence of a control group limits our ability to directly compare the effectiveness of virtual consultations to in-person visits, we were able to identify significant differences in UPDRS measurements between the in-person visit and virtual visit periods. These findings suggest that the pandemic and the associated shift to virtual care had an impact on the progression of Parkinson's disease symptoms.

To clarify our methodology, we have revised the relevant section in the methods to provide a clearer explanation of our comparative approach (lines 134-157). Furthermore, in the discussion section, we have delved deeper into the implications of these findings, highlighting the potential influence of the pandemic lockdown on symptom changes (lines 236-273).

We appreciate the reviewer's comment, which has prompted us to provide a more comprehensive explanation of our methodology and its limitations.

References:

  1. Schrag, Anette, et al. "Rate of clinical progression in Parkinson's disease. A prospective study." Movement disorders 22.7 (2007): 938-945.
  2. Bugalho, Paulo, et al. "Progression in parkinson's disease: variation in motor and nonmotor symptoms severity and predictors of decline in cognition, motor function, disability, and healthrelated quality of life as assessed by two different methods." Movement Disorders Clinical Practice 8.6 (2021): 885-895.
  3. Horváth, Krisztina, et al. "Minimal clinically important difference on the Motor Examination part of MDS-UPDRS." Parkinsonism & related disorders 21.12 (2015): 1421-1426.

Comment 3:

The Manuscript is not well written, some sections such as Results and Figures are not clear to understand due to absence of any statistical description.

In the Material and Methods patient section Authors should indicate mean age ± standard deviation of patients and the number of men and women enrolled in the study.

In addition, Authors should include in the Material and Methods a section of statistical analyses to describe the type of statistical analyses and the software used.

In the Results section the figure 3 is not clear because it is not described the p values for each asterisk. The representation of statistical differences in the Figure 3 should be described such as

*P<0.05, **P<0.01 and < 0.001**. In the legend of the Figure 3 Authors described only acronyms without any statistical description and information.

Response 3:

We appreciate the reviewer's feedback regarding the clarity and statistical description of our manuscript. We have taken the following steps to address these concerns:

Material and Methods: We have included the mean age ± standard deviation of patients and the number of men and women enrolled in the study in Table 1. This table provides a comprehensive overview of the demographic characteristics of the study participants.

Statistical Analyses: We have added a subsection in the Material and Methods specifically dedicated to describing the statistical analyses conducted in our study. This subsection provides information on the type of statistical analyses performed and the software used for data analysis. By including this information, we aim to enhance the transparency and reproducibility of our study.

Results Section and Figure 3: We have revised the figure captions and provided clearer statistical descriptions for Figure 3. Specifically, we have added the p-values corresponding to each asterisk to indicate the level of statistical significance. The representation of statistical differences in Figure 3 is now described as *P<0.05, **P<0.01, and ***P<0.001. We have also updated the figure legend to include the necessary statistical information. These changes ensure that the reader can better understand the significance of the observed differences (Figure 3 and related captions).

---

We are grateful for the reviewer's insightful comments, which have prompted us to enhance the discussion of the study's limitations and potential avenues for future research.

Round 2

Reviewer 1 Report

Comments for authors

Brain Sci. – Manuscript ID: brainsci-2478693 (R1) – “Remote Assessment of Parkinson's Disease Patients Amidst the 2 COVID-19 Lockdown in Mexico.”, by Rodrigo León-García, Emmanuel Ortega-Robles and Oscar Arias-Carrión.

The authors submitted a carefully-prepared revision, which satisfactorily addressed the remaining concerns.

They have addressed all comments.

Reviewer 2 Report

The quality of the paper work has overall improved. Please pay attention to the use of abbreviations along the manuscript and keep them consistent (always use the same abbreviation to indicate the MDS-UPDRS). 

Reviewer 3 Report

I appreciate the revised draft, however  the Manuscript is partially improved, because  the main limitations of this study such as the low number of subjects enrolled and the absence of a control group are not solved. I think that the reasearch is not correctly conducted and additional experiments are needed to solve these two main problems.

Moderate editing of English language is required